# Glymphatic Dysfunction Induced Oxidative Stress and Neuro-Inflammation in Major Depression Disorders

**DOI:** 10.3390/antiox11112296

**Published:** 2022-11-20

**Authors:** Simeng Gu, Yumeng Li, Yao Jiang, Jason H. Huang, Fushun Wang

**Affiliations:** 1Department of Neurology, The Affiliated Hospital of Jiangsu University, Jiangsu University, Zhenjiang 212001, China; 2Department of Psychology, Medical School, Jiangsu University, Zhenjiang 210023, China; 3Institute of Brain and Psychological Sciences, Sichuan Normal University, Chengdu 610066, China; 4Department of Neurosurgery, Baylor Scott & White Health, Temple, TX 79409, USA; 5Department of Surgery, College of Medicine, Texas A & M University, Temple, TX 79409, USA

**Keywords:** glymphatic system, astrocyte, microglia, inflammation, major depressive disorder, reactive oxygen species

## Abstract

Major Depression disorder (MDD) is a potentially life-threatening mental illness, however, many patients have a poor response to current treatments. Recent studies have suggested that stress- or trauma-induced oxidative stress and inflammation could be important factors involved in the development of MDD, but the mechanisms remain unclear. We showed that the glymphatic system is a recently discovered structure in the brain that may be involved in the clearance of large molecular and cell debris in extracellular space. In addition, the glymphatic system can help with the removal of reactive oxygen species (ROS) and cytokines such as IL-1β and HIF-1α. Glymphatic impairment can lead to ROS accumulation in the microenvironment, inducing cellular injury signaling and activating NLRP3 in microglia to induce inflammation and, thus, many brain diseases, including psychiatric disorders. Therefore, trauma-induced glymphatic impairment could induce oxidative stress and inflammation, and thus MDD. This paper will review recent advances with regard to stress-induced glymphatic system impairment and ROS-mediated inflammation in MDD.

## 1. Introduction

Major depressive disorder (MDD) is a prevalent emotional disorder characterized by a loss of hedonic motivation, cognitive and behavioral retardation, and a susceptibility to commit suicide; however, its etiology remains unclear [1]. Increasingly, further possible mechanisms of MDD are emerging, such as neurotransmission alternation, neurotrophic changes, neuroendocrine changes including the HPA axis, inflammation, nutrition, and the brain-gut axis [2]. Recent studies have suggested that oxidative stress and inflammation may be the main causes of MDD [3,4]. Oxidative stress is caused by a homeostatic impairment-induced imbalance between antioxidants and reactive oxygen species (ROS), which can lead to DNA, proteins, or lipid damage. ROS can modulate neurotransmitter (such as monoamine) release and monoamine oxidase activity, an enzyme that metabolizes monoamine, such as dopamine (DA), serotonin (5-HT), norepinephrine (NE), which in turn can enhance ROS production in mitochondria [5].

The brain utilizes approximately 20% of the total energy and oxygen supply of the body but only weighs approximately 2% of the body, thus, energy supply is critically important for the brain [6]. Due to its high energy requirement and high lipid content, the brain is especially vulnerable to ROS. In addition, the neurons are well equipped with many more mitochondria to produce ATP, which is the only source of energy for neurons. Evolutionarily, ROS is a biological product of mitochondria that antagonizes microorganisms and has evolved with cellular signaling abilities, such as pro-inflammatory signaling [7]. Too much ROS in the extracellular space has been proven to be the main cause of neurodegeneration, and is unequivocally established as being involved in the pathogenesis of MDD [8]. Previously, we have shown that ROS overload impairs adult neurogenesis, neuro-inflammation, and neurodegeneration, and targeting ROS may provide a novel way to combat various brain diseases [9].

The glymphatic system is a recently discovered network that works to maintain homeostasis in the microenvironment by enabling fluid exchange between cerebrospinal fluid (CSF) and interstitial fluid (ISF). Before the glymphatic system theory was established in 2012, it was assumed that the brain recycled most of its protein waste [10]. However, the introduction of the glymphatic system has changed this concept, and the glymphatic system is now regarded as a brain-wide perivascular pathway for waste clearance of metabolite products from the brain [11,12]. The glymphatic system is currently known to play a critical role in many brain disorders [13]. Stressed brains carry ROS, an inflammatory component, which is likely to promote susceptibility to depression. Thus, controlling ROS and inflammation could be a good way to treat MDD [14], or ROS and inflammatory processes might be a major cause of MDD, and it is really the case that ROS and the immune system regulate mood in MDD.

In this paper, we reviewed recent studies about the glymphatic system, and proposed a new perspective for its function in MDD. Firstly, we introduced the definition of the glymphatic system and its major functions in waste clearance in the brain, including ROS, which might play a pivot role in MDD; secondly, we explored the roles of ROS in MDD, and introduced two major ways of modulating ROS: generation of ROS in mitochondria and antioxidant supplementation in the gut-brain axis. Then we compared these processes with the traditional monoamine hypothesis. At the end, we reviewed the functions of two major glial cells in modulating the glymphatic systems and inflammation process. We hope this review will add new and impactful perspective for future studies of the glymphatic system and MDD, which might shed light on the mechanisms and treatment for MDD.

## 2. The Glymphatic System

The lymphatic system plays a pivotal role in controlling inflammation; however, it has been assumed that the brain has no similar lymphatic network. Recently, the glymphatic system has been found to play a similar role in neuroinflammation in the brain, as well as in the maintenance of brain homeostasis [11], similar to the lymphatic system in the peripheral tissue [15]. In addition, we have found that the glymphatic system works together with the meningeal lymphatic vessel to accomplish this function [16]. Recently, Louveau et al. defined meningeal lymphatic vessels with immunofluorescence staining [17]. The glymphatic system is currently regarded as an important circulation supplementary system, regulating homeostasis by removing the substances produced by cell death or metabolism, and inducing immune reactions [18]. The discovery of the cerebral glymphatic system has provided a revolutionary perspective elucidating the pathophysiological mechanisms of many brain disorders.

The glymphatic system is different from the other lymphatic systems in that it is composed of astrocytes, while the other lymphatic system consists of endothelial cells (Figure 1). Astrocytes have long been regarded as “housekeeping cells” for maintaining cerebral homeostasis. However, the past three decades have witnessed an understanding of the active function of astrocytes to actively regulate blood flow, provide an energy supply to neurons, modulate the blood-brain barrier, remove metabolite waste and regulate homeostasis in extracellular space in the brain, including in fluids, ions, and neurotransmitters [19]. An increasing number of studies have further highlighted the contribution of astrocytes to actively removing metabolite waste by forming the unique glymphatic system structure [20]. So, unlike lymphatic vessels in other organs, the glymphatic system is composed only of perivascular spaces and astrocytes (Figure 1), however, it has a similar structure for effectively controlling the clearance of toxic materials from the microenvironment [21], and its dysfunction may contribute to MDD [22].

### 2.1. Clearance Function of the Glymphatic System

The glymphatic system is a unique drainage system that can drain fluid together with many other substances, such as macromolecules, ROS, cytokines, and antigens, from interstitial fluid to cerebrospinal fluid and finally to meningeal lymphatic vessels, to remove these materials to the peripheral lymphatic system [23]. The typical clearance materials include the following.

The glymphatic system in tau clearance: the glymphatic system helps the clearance of macro-molecules such as amyloid-β and tau, which may be a major cause of many brain disorders. There are numerous factors that are involved in many brain disorders such as oxidative stress, neuroinflammation, and apoptosis, which could be the reason for the accumulation of extracellular amyloid-beta peptides [24]. It has been found that tauopathy in animals is caused by an impaired glymphatic system [25], and neuroinflammation, cytokines, chemokines, and the complement system also play a major role in MDD [26].

The glymphatic system in K^+^ clearance: The glymphatic system is also involved in extracellular potassium (K^+^) buffering and in the regulation of extracellular space [27] (Figure 1). High extracellular K^+^ plays an important role in modulating extracellular space during pathological and physiological conditions, such as edema or seizures. Astrocytes have a key function in K^+^ clearance via two processes: K^+^ uptake and K^+^ spatial buffering, which depend on Kir channels and the Na^+^/K^+^-ATPase [28]. Recently, it was found that the glymphatic system is involved in controlling extracellular K^+^ [29,30].

The glymphatic system and ROS clearance: reactive oxygen species (ROS) can exert detrimental effects at a high level or beneficial effects at lower levels [31]. For example, our previous study found that ROS may exert neuro-generative effects via modulating growth factors [9]. In contrast, oxidative stress due to enhanced ROS generation and/or reduced levels of antioxidants can induce cell apoptosis and death [32]. The glymphatic system is the major physiological clearance pathway for ROS and ROS-related inflammation in neurodegenerative diseases [33].

The glymphatic system and cytokines: Recent studies have further highlighted the part played by the glymphatic system in inflammation via the removal of pro-inflammatory cytokines and chemokines, such as tumor necrosis factor-α (TNF-α) and interleukin-1β (IL-1β) [34] or HIF-1α [35]. In addition, NLRP3 inflammasome activity can also be affected by the glymphatic system via the modulation of extracellular ATP released from astrocytes. Our previous studies also suggested that biberine could affect astrocytic activity and possibly the glymphatic system to help reduce cytokine production and release, and may be a potential mechanism for the treatment of seizures [36].

### 2.2. Glymphatic System Depends on the Polarization of AQP4

The normal function of the glymphatic system depends on the polarized distribution of AQP4 [15]. CNS is enriched in astrocyte end feet, where water exchange and extracellular space are controlled [37]. Furthermore, it has been found that a decreased polarization of AQP4 reduces CNS edema [38]. Defects in the glymphatic system altogether may lead to chronic neurodegeneration and tauopathy in old age in IL33-deficient mice [39].

AQP4 is a kind of water channel that is involved in water movement across the cell membrane. Many studies have proved that AQP4 is a major driver for the influx of cerebrospinal fluid (CSF), depending upon the perivascular [40]. For example, several experiments with AQP4 knockout (Snta1 KO) found that the CSF influx is lower in these mice [38], and the brain edema formation is slower, suggesting that aquaporin-4 (AQP4) is a primary influx route for water during edema formation [37]. In addition, Thrane et al. found that AQP4 deletion can reverse hypo-osmotic stress (20% reduction in osmolality)–induced astrocytic Ca^2+^ spikes [41]. These studies suggest that AQP4 not only acts as an influx route for physiological water transportation, as well as downstream pathways, such as inflammations, which may exacerbate the pathological conditions associated with brain diseases such as MDD [41].

In addition, AQP4 was found to regulate monoamine neurotransmission; for example, one study found that AQP4 knockout mice have high levels of monoamine, including 5-HT and NE in the medial prefrontal cortex [42]. Interestingly, the interaction is affected by sex, as the DA and 5-HT levels were significantly increased in the prefrontal cortex in the male AQP4 knockout mice, but not in the female mice. These results suggested that AQP4-induced water influx affects monoamine metabolism in a region-specific way [42].

## 3. ROS and Inflammation

Dysfunction of the glymphatic system can induce ROS accumulation and pro-inflammatory signaling, damage vital macromolecules, and induce cellular apoptosis during MDD [43]. Many recent studies have found that an increased generation of ROS and induced inflammation-caused MDD [44]. The brain is vulnerable to oxidative stress, mostly due to its high oxygen consumption, high lipid content, and weak anti-oxidative defenses [8]. ROS can activate inflammasomes in microglia, and produce inflammation cytokines, including TNF-α, IL-1β, and IFN-γ [45,46]. Inflammation can impair neuroendocrine-immune functions and lead to many infectious diseases, such as MDD [47]. Pro-inflammatory cytokines have emerged as pathological biomarkers of MDD, and an effective strategy for treating MDD may be to use suitable antioxidants to antagonize ROS.

Even though monoamine deficiency is the most well-known pathophysiological mechanism of MDD, many studies have proved the critical role of inflammation in MDD. A number of recent papers have suggested that pro-inflammatory cytokines, including tumor necrosis factor-alpha (TNF-α), interferon-gamma (INF-γ), and interleukin-6 (IL-6) induced by oxidase stress, may contribute to MDD [46,47,48]. However, inflammatory cytokines work together with monoamine transmission in MDD; thus, we present an integrative hypothesis that integrates monoamine transmission with immunological alterations in MDD (Figure 2). Furthermore, it is hypothesized that chronic unpredictable mild stress (CUMS)-induced glymphatic pathway dysfunction might act as a bridge between inflammation and monoamine disturbance in MDD [23,49]. Monoamine is the major neurotransmitter for waking and sleep, which also affects the glymphatic system by making it work more efficiently during sleep, and blocking its function via NE release during arousal [50]. Recently, it has been found that CUMS-induced NE release can decrease the polarization of AQP4 expression in astrocytes, inhibiting the function of the glymphatic system and inducing oxidative stress and inflammation, leading to depression-like symptoms [51].

### 3.1. Reactive Oxygen Species

Reactive oxygen species (ROS) are free radicals that are composed of singlet oxygen, superoxide, hydrogen peroxide, or hydroxyl radical molecules [52]. ROS is mostly produced in mitochondria as byproducts of cellular metabolism when creating energy from food and oxygen. In mitochondria, the electron transport chain is not only the mechanism of ATP, but it is also the mechanism for the production of ROS-like superoxide radicals. ROS are enzymatically degraded by manganese superoxide dismutase (SOD) as well as glutathione, and by glutathione peroxidase, catalase, and peroxiredoxin outside the mitochondria, which acts as an antioxidant mechanism. Nevertheless, oxidative stress can be induced by ROS overproduction, particularly in the brain, where mitochondrial activity is the most active. Oxidative stress can also be induced by a reduced antioxidant supply, together with an increased generation of ROS in mitochondria. Thus, elaborate antioxidant defense systems, such as superoxide dismutase (SOD), are needed to minimize the levels of ROS, and a failure to maintain redox homeostasis will result in inflammation, leading to cell necrosis [53].

### 3.2. Brain-Gut Axis and ROS

In addition to impaired ROS scavenging and excessive ROS generation, oxidative stress can also be affected by reduced antioxidant supply in the gut. It is found that low levels of antioxidant compounds such as vitamins C and E, and co-enzyme Q-10 [54]. Thus, antioxidant dietary supplements may provide beneficial effects for MDD patients. In fact, a lack of biological antioxidants has only recently been proposed as a reason for MDD [55,56,57]. Antioxidants are oxidizable substrates from fruit and vegetables that are absorbed in the gut and are possibly affected by gut microbiota [57].

The role of diet in MDD has recently been explored in some studies, where it was shown that antioxidant diets and anti-inflammatory may help in the treatment of MDD [58]. For example, a recent study showed that an omega-3 polyunsaturated fatty acid (PUFA)-enhanced diet reduced the symptoms of MDD. However, the improper use of dietary supplements may lead to anti-oxidative stress; for example, some recent studies showed that antioxidant therapy had no effect and even increased mortality. This was hypothesized as being due to the oxidative stress compensative model, which suggests that the supply of only one antioxidant might increase oxidative stress [53].

Nevertheless, many studies suggested that the gut microbiota has a critical role in supplementing bacteria-derived metabolites [59], such as polyunsaturated fatty acids [60], thus playing an important role in MDD by modulating homeostasis [61]. Indeed, the gut microbiota is suggested to be a potential tool in helping to protect against MDD by supplementing unsaturated fatty acids, monoamine neurotransmission, neuroendocrine pathways, and inflammation [62,63]. In contrast, the gut microbiome might supply cytokines, such as inflammation factor interferon-γ (IFNγ), affecting astrocyte function and inducing MDD [64]. Astrocytes exert important functions in homeostasis and in the pathogenic activities of MDD, along with activating microglia to induce inflammatory reactions in MDD [64].

### 3.3. ROS, Glia, and Inflammation

Oxidative stress and an excessive accumulation of ROS can induce the activation of inflammasomes in the microglia [65]. The overproduction of inflammatory cytokines has been proven to play important roles in MDD [66,67]. Microglia are the endogenous immune cells in the brain that detect CNS homeostasis disorder, and respond by secreting inflammatory cytokines and chemokines [68]. Microglia activation induces NLRP3 inflammasome overexpression and the release of many cytokines [69]. Over the last decade, accumulating evidence has shown that microglia impairment can induce MDD [70].

Activated microglia release inflammatory factors such as TNF-α, IL-1α, and PGE2, which can activate and thus affect clearance by glymphatic systems, which can, in turn, modulate extracellular ROS and cytokines. The cross-signaling between microglia and astrocytes is crucial in determining the intensity and timing of neuroinflammatory reactions [49]. In addition, ROS and inflammatory cytokines can work together to affect monoamine neurotransmitters and activate microglial cells [71,72].

## 4. Monoamines and the Glymphatic System

Monoamine deficiency was previously suggested as the major cause of MDD, and most antidepressants are used to increase the amount of monoamine neurotransmitters [73]. However, this might be an outdated explanation for the cause of MDD and needs to be revised to reflect the more complex changes in the brain associated with the illness. While we do not fully understand how antidepressants work, we know that increasing monoamine neurotransmitter is an oversimplification. For example, monoaminergic antidepressants are not effective in some difficult-to-treat patients, working too slowly to be effective, especially the 5-HT—specific antidepressants. In addition, although many studies have confirmed the critical roles of monoamine in MDD, their effects are still unclear; and the difference between the three monoamines (NE, DA, and 5-HT) in MDD is not clear. Previously, we originally proposed the “Three Primary Color Model of Basic Emotions”, trying to differentiate the functions of these three monoamines. We hypothesized that the three monoamines work differently to make three distinct emotions, as in the three primary colors [74,75]. Here, we further propose that their effects in treating MDD are also different in that catecholamine can directly induce emotional arousal, while 5-HT can induce sedation and sleep. So, the effects of 5-HT are enhancing clearance via the glymphatic system and benefiting MDD (Figure 3).

Monoamine neurotransmitters are perfectly suited to the regulation of mitochondria in the brain, and enhancing the energy supply function, and mitochondria have recently been found to play a pivotal role in emotional arousal. Emotion arousal can induce NE release in the locus coeruleus and also in other monoamine neurotransmitters [75]. However, a side-effect of monoamine release is enhancing neural activity through the parallel activation of both neuronal mitochondria and astrocytic clearance to establish a synergistic mechanism of antagonizing ROS-induced oxidative stress and inflammation. It is found that chronic stress can activate excessive monoamine oxidase activity, and induce mitochondrial oxidative stress via excessive ROS generation in the mitochondria. Monoamine neurotransmitters can also directly affect the clearance function of the glymphatic system [76]. For example, previous studies in our laboratory suggested that monoamines may affect active astrocytic spatial buffering ability via the enhanced activity of the Na^+^ pump [77]. The glymphatic system is involved in metabolic conversion of ROS and inflammation, which could contribute to both ROS accumulation and inflammatory reaction, as well as monoamine deficiency and glutamatergic hyper-function [78].

The effect of NE on the glymphatic system: The glymphatic system is regulated by sleep and NE, with excessive NE inhibiting fluid movement [79]. Chronic stress is a risk factor for depression in humans and animal models, and it is a typical model in the study of the neurobehavioral alterations relevant to depression. NE is known as the neurotransmitter for stress and emotional arousal [80,81], inducing “fear and anger” emotions [74] and “fight or flight” behavior (Figure 3) [82]. NE plays a pivotal role in emotional arousal and in antagonizing MDD, but too much NE and cortisol induced by chronic stress, such as chronic neuropathic pain, can induce sleep disruption and depression [47]. In addition, excessive NE-induced glial activation in both microglia and astrocytes, can negatively affect the function of glymphatic system [83,84]; furthermore, excessive NE robustly inducing the generation of ROS, and antioxidants, such as dexmedetomidine [85,86].

Effect of DA on the glymphatic system: MDD is characterized by the lack of positive emotion (i.e., anhedonia), and the symptoms in rodent models of MDD include reduced reward-seeking behavior and less stressed struggling behavior. Anhedonia is associated with DA system dysfunction in both humans and rodents; while less stressed struggling behavior is related to a lower NE function (Figure 3). Unlike traditional antidepressants, the new drug ketamine, which is a type of stimulant, can rapidly alleviate depressive symptoms in MDD via actions on the DA system. Thus, raising the level of brain DA can induce emotional arousal, and change is a quick way to treat depressed mood [87]. The major mechanism by which ketamine exerts its antidepressant effect is related to DA release [88]. Thus, astrocyte-targeted DA might be a therapy intervention designed for MDD [89], and catecholamines can induce emotional arousal and activate the fundamental functions of astrocytes [90].

Effect of 5-HT on the glymphatic system: Monoamine neurotransmission alternation, especially 5-HT, has proved to be the most significant pathophysiological etiology for MDD in the last century [91], and the first-line treatment of MDD still targets the serotonin system [92]. However, the effects of 5-HT on the brain are somewhat controversial as it has 14 receptors with different functions. For example, both in vitro and in vivo studies have demonstrated that the 5-HT receptor 4 (5-HT4 R) can inhibit glutamatergic synaptic transmission [93].

Astrocytes can modulate neural networks via purinergic pathways, cortisol, and 5-HT, which may work on cortical inhibition via the GABA and purinergic pathways [77]. Many studies have suggested that astrocytes constitute effectors of the 5HT-mediated decrease in frequency transmission via enhancing the cortical inhibitory tone [94]. This is consistent with previous studies suggesting that the major function of 5-HT is sedation and inhibiting suicidal ideation; consistently, the real function of central 5-HT in treating MDD may be in inducing sleep and sedation, and in inhibiting compulsive thoughts [95,96]. Serotonin (5-HT) is both a neurotransmitter in the CNS and a paracrine and endocrine signal in the gut, where it can control the feeding behavior of animals. Thus, the function of 5-HT may be as discussed (Figure 3); and 5-HT can induce the behavioral inhibition process, which is similar to the prolonged helpless state in MDD [97]. In fact, many recent studies have focused on the inhibitory effects of 5-HT [98]. In addition, 5-HT can also be converted into melatonin by endocrine to induce anti-stress responses in the brain, and to affect sleep [99].

Effect of sleep on the glymphatic system: Sleep is evolutionarily conserved in the mammal species, and impaired sleep is a common trait in many disorders. Sleep is modulated by monoamine neurotransmitters, with 5-HT released from the raphe nuclei to induce sleep, while NE in the locus cereus induces waking. Sleep can help with the housekeeping recovery of the brain from stress or arousal, with energy supply, and with waste clearance. The glymphatic system is mostly active during sleep, while excessive catecholamine can diminish the glymphatic system [21]; really, CUMS (chronic unpredictable mild stress) can impair the glymphatic system and induce MDD. However, acute stress and emotional arousal are needed for MDD, as MDD patients require emotional arousal to treat depressive states. Previously, we summarized the roles of monoamine in emotional arousal and proposed their roles in MDD, originally presenting the different roles of monoamine in the three core affects: reward (DA), punishment (5-HT), and stress (NE) (Figure 3) [80].

Monoamine, possibly together with induced emotional arousal, can activate astrocytic activity. The integrative pathway might be that the glymphatic system and related glial cells work as a housekeeping system that helps the normal brain function, while excessive monoamine antagonizes the glymphatic system and those housekeeping effects. Indeed, serotonin treatment has been found to significantly reduce ROS formation [100]. In other words, long-term stress and excessive monoamine release impairs the glymphatic system and leads to ROS accumulation, and sleep is needed to help the glymphatic system clear up the brain and recover from CUMS (Figure 3).

## 5. Astrocytic Ion Channels and the Glymphatic System

Determining the neurobiological mechanisms of MDD is an active field, and many studies have suggested that astrocytes play a central role in MDD, and MDD is characterized by depressed astrocytic activity, reduced astrocyte numbers, and smaller size; unlike other diseases. The reduced function of astrocytic activity can impair the glymphatic system, which might be a major pathway for MDD [101]. The impairment of astrocytic function includes many ion channels, such as Na^+^/Ca^2+^ exchanger, Na^+^/K^+^/2Cl-cotransporter 1, as well as glial fibrillary acidic protein (GFAP), and aquaporin-4 (AQP4) [102,103].

NCX: Na^+^/Ca^2+^ exchanger (NCX) plays a pivot role in the potentiation of Ca^2+^ entry in astrocytes [28], Ca^2+^ entry via activated NCX together with ROS-induced dysfunction results in dysfunction of the glymphatic system [104], and the inhibition of Ca^2+^ overload or closing NCX could reverse ROS increase and cause cell injury [104]. NCX-dependent signaling can be activated by gliotransmitters 5-HT, ATP, and ADP, and thus decrease the accumulation of ROS and cytokines [105]; contrarily, the activation of dopaminergic or adrenergic receptors enhances the Na^+^ pump, causing an increase in mitochondrial ROS levels [106,107]. In contrast, ROS can also affect NCX-induced Ca^2+^ entry [108].

NKCC: Extracellular space, which is an important part of the glymphatic system, is modulated by extracellular K^+^ via mechanisms that involve passive spatial buffering mediated by Kir4.1 and Na^+^/K^+^/2Cl-cotransporter 1 (NKCC1) in astrocytes, and/or active transporting via Na^+^/K^+^-ATPase activity. In addition, the excess K^+^ can also affect the glymphatic system by changing the extracellular space. It has been found that NKCC1 works with a potential role in K^+^ clearance and the extracellular space’s associated shrinkage [109]. Thus, NKCC1, Kir4.1, and Na^+^/K^+^-ATPase might work together to modulate the clearance of extracellular K^+^ transients, as well as ROS [109]. The glia cell swelling and extracellular space dynamics are associated with K^+^ uptake and/or water intake through AQP4; recent studies have also suggested a relationship with K^+^-mediated glial depolarization and metabolic demand [110].

In all, MDD is a common emotional disorder that seriously affects people’s quality of life, however, the molecular mechanisms are complex due to its clinical and etiological heterogeneity. Recent studies suggested that astrocytes play a central role in the etiology of MDD.

## 6. Glymphatic System and Microglia

In addition to the astrocytes’ dysfunction, over-activated microglia can induce neuroinflammation in the hippocampus of stress model mice, inducing depressive-like behaviors. Microglia are the only endogenous immune cells of the brain, and activated microglia can activate the NLRP3 inflammasome, which is a major player in mediating neuroinflammation (Figure 4). The NLRP3 inflammasome, in turn, induces the transformation of microglia from a resting state to a pro-inflammatory state to release pro-inflammatory cytokines. In contrast, inhibiting microglia from reducing inflammation is possibly one of the main pathogenic mechanisms of MDD [111].

Our previous studies suggested that oxidative stress overload can lead to neuroinflammation and many brain disorders [9]. Glymphatic system dysfunction has been proven to be a major cause of neurodegenerative diseases (NDs) that result from a reduction in the clearance of ROS and inflammation. The glymphatic system through aquaporin 4 (AQP4) in astrocytes and perivascular space to promote the water flow between the interstitial fluid (ISF) and CSF. An excessive accumulation of water can lead to edema. Changes in the glymphatic system may be an important factor for MDD [112].

MDD is a serious emotional disorder characterized by anhedonia and a loss of energy as reported in the DSM-IV stipulation [113], which is typically shown as a “depressed mood” and “loss of interest”, as well as symptoms of fatigue, sleep disturbance, and sexual dysfunctions”. However, the main molecule pathological features of depression include extensive neuroinflammation. Therefore, NLRP3 inflammasome-mediated microglia activation, and the secretion and release of TNF-α, IL-1α, and PGE2, may be major causes of depression-like behavior (Figure 4). The glymphatic system is a highly polarized cerebrospinal fluid (CSF) transport system that facilitates the clearance of neurotoxic molecules through a brain-wide network of perivascular pathways [114]. The schema in Figure 4 shows the relationships among the microglia, ROS, and cytokines in perivascular CSF transport [115]. In all, the core symptoms of MDD involve a loss of interest or pleasure, tiredness and lack of energy, reduced appetite, and weight loss, as well as a feeling of worthlessness or self-blame [116]. The cause of MDD is due to inflammation which is induced by the low antioxidant capacity and high oxidative damage.

## 7. Conclusions

This review summarized recent studies about astrocytes, microglia, and inflammasome pathways that are activated during the occurrence and development of MDD, and provided a novel perspective for the mechanism of MDD. The perivascular space surrounding small blood vessels has recently been defined as the glymphatic system, providing a revolutionary perspective on the many pathophysiological mechanisms of MDD. Research into the function and pathogenesis of this system has proved that it carries out similar functions to the lymphatic system in the body, playing an important role in removing metabolic waste and maintaining homeostatic fluid circulation in the brain. In this article, we briefly described the factors influencing the cerebral glymphatic system, their effects on ROS accumulation and inflammation, and their roles in MDD. The aim of this research was to provide a perspective for future studies of the glymphatic system and MDD, and is expected to provide a new etiology and neural basis for the mechanisms and treatment for MDD.

## Figures and Tables

**Figure 1 antioxidants-11-02296-f001:**
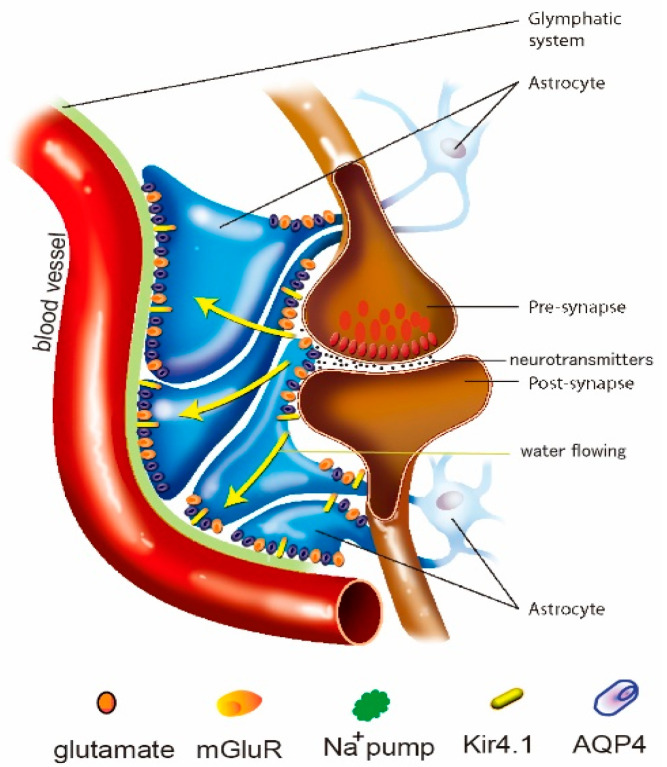
Schematic drawing showing the structure of the glymphatic system, which is composed of the perivascular space (PVS) and astrocytic end feet. The fluid passes from interstitial fluid to cerebrospinal fluid (CSF) in the PVS around the small blood vessels in the brain, which is important for the clearance of ROS and cytokines, etc. The glymphatic system can actively transport the fluid through ion channels, such as AQP4 (purple dots), K^+^ channels (such as NKCC1; orange dots), and co-transporters (such as Glt1; yellow dots).

**Figure 2 antioxidants-11-02296-f002:**
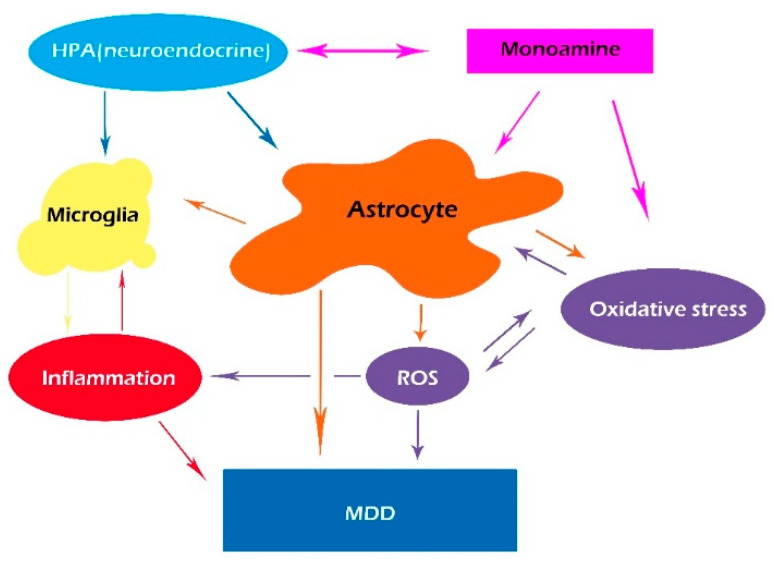
An integrative hypothesis suggesting that the glymphatic system integrates monoamine transmission with dysregulation in neuroendocrine systems, together with oxidative stress and induced immunological alterations in MDD.

**Figure 3 antioxidants-11-02296-f003:**
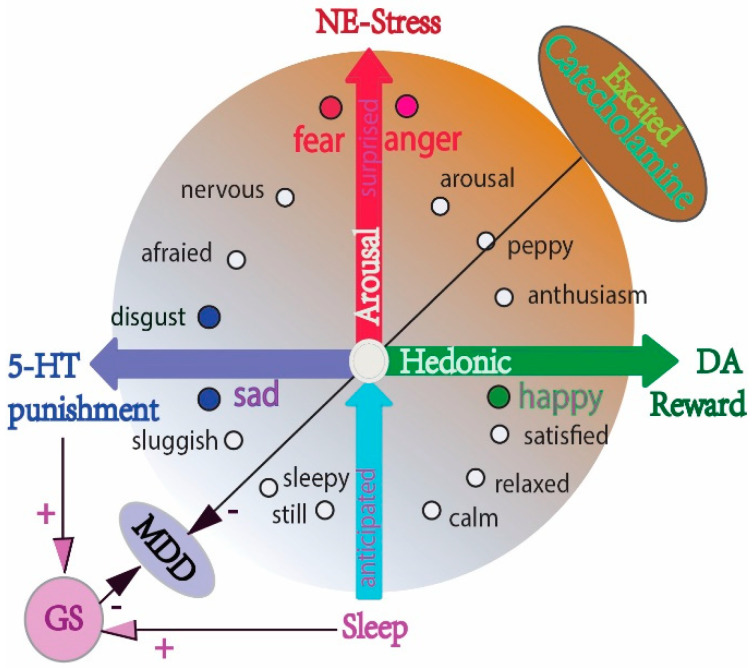
The different effects of monoamine on emotional arousal and sleep, on the glymphatic system and on MDD. The schema shows the relationship between monoamine and basic emotions (DA: reward, NE: stress, 5-HT: punishment) (modified from our previous paper [74,75]), and the two-dimensional model of emotions (valence and arousal). The major effects of 5-HT may be sedation and the inhibition of suicide ideation. Excessive catecholamine (DA and NE) antagonize the glymphatic system, while serotonin (5-HT) facilitates sleep and helps the glymphatic system with its housekeeping function. Chronic stress and induced excessive catecholamine release impair the glymphatic system, leading to ROS accumulation and MDD.

**Figure 4 antioxidants-11-02296-f004:**
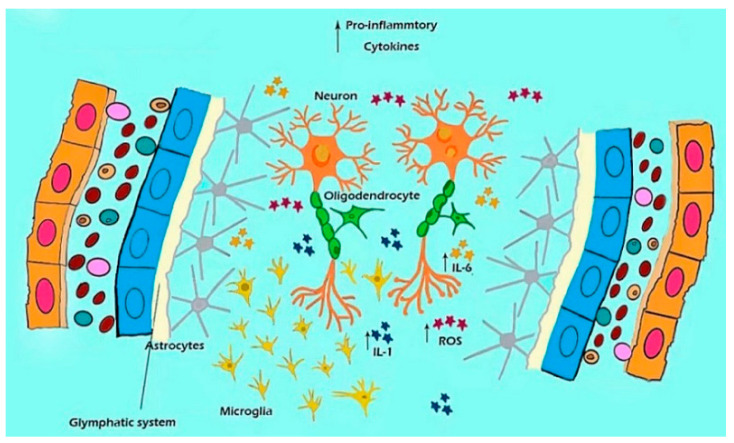
A model shows the contribution of microglial to the dysfunction of the glymphatic system, inducing ROS accumulation and inflammation in MDD. ROS accumulation and neuro-inflammation are the major reasons for MDD.

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
