# Peer review of "Glymphatic Dysfunction Induced Oxidative Stress and Neuro-Inflammation in Major Depression Disorders"

_antioxidants, 2022, doi:10.3390/antiox11112296_

Round 1
Reviewer 1 Report
Dear Authors,
the topic of the manuscript is of great interest and novel. Pictures are clear and appreciable.
However, in my opinion it would need an extensive editing of english language, beside scientific reliability.
Please find possible suggestions from the abstract:
Major Ddepression disorder (MDD) is a potentially life-threatening emotional illness, but CURRENT the treatmentS is ARE OFTEN not effective. Recent studies suggested that stress or trauma-induced oxidative stress and inflammation are COULD BE important factors for INVOLVED IN the development of MDD, but the mechanismS is REMAINS not UNclear.
WE RECENTLY SHOWED THAT IN THE BRAIN THE Glymphatic system is a recently found structure (?) in the brain in our lab, which might be involved in THE clearance of large molecular and cell debris in extracellular space.
Author Response
Dear Authors,
The topic of the manuscript is of great interest and novel. Pictures are clear and appreciable.
Response: Thank you very much for the positive comments.
However, in my opinion it would need an extensive editing of English language, beside scientific reliability.
Please find possible suggestions from the abstract:
Major Ddepression disorder (MDD) is a potentially life-threatening emotional illness, but CURRENT the treatmentS is ARE OFTEN not effective. Recent studies suggested that stress or trauma-induced oxidative stress and inflammation are COULD BE important factors for INVOLVED IN the development of MDD, but the mechanismS is REMAINS not UNclear.
WE RECENTLY SHOWED THAT IN THE BRAIN THE Glymphatic system is a recently found structure (?) in the brain in our lab, which might be involved in THE clearance of large molecular and cell debris in extracellular space.
Response: Thank you very much for the valuable suggestions, we have made fundamental changes in the language by a native English speaker and also by the MDPI publishing agency. Please see the changes in the text.
Reviewer 2 Report
This review provides an important update on neuroinflammatory mechanisms that contribute to the development of major depressive disorder (MDD), an illness that is a significant contributor to the overall global burden of disease. Work from the authors and others have made important advances in explaining how oxidative stress, inflammation, and alterations in the the gymphatic system are invovled in the pathophysiology of the illness. The review is well-organized and highlights the relevant structures, cells, molecules and mechanisms, with the figures providing additonal explanations and clarifications.
Below are the issues I recommend the authors address prior to publication:
The primary issue with this paper is with English language and style. Extensive revisions are needed throughout the paper to improve English grammar and usage. I will provide some general and specific examples below, but most paragraphs need one or more revisions.
Inconsistent use of singular vs. pluaral
Present and past tense in the same sentence
"the" is missing in several sentences, for example:
THE Glymphatic system is a recently found structure….. (line 16)
More and more possible mechanisms for MDD are emerging... (line 31) "More and more" is not appropriate
Too much use of "Indeed"
Do not use only the abbreviation MDD in the title (at least spell it out in the title).
With regard to content I recommend reviewing sections of the manuscript that disucss clinical aspects of MDD and treatment. Sentences that describe the symptoms of depression could be more complete. As a psychiatrist reading this draft, my impression is that the authors don't fully understand the clinical presentation of the illness or what we currently understand about treatments. I suspect this is becuase of how the paper is written and the English language issues rather than any lack of knowledge on the part of the authors. For example, saying in line 476 that "MDD is a serious emotional disorder characterized by anhedonia, loss of energy" is incomplete. Revising this statment to include a more complete description of the symptoms of depression is needed.
I also disagree with the sentence in line 271 "Monoamine deficiency has been suggested to be the major reason for MDD, and most antidepressants are used to increase the monoamines neurotransmitters." This is an outdated explanation for the casue of MDD and needs to be revised to reflect the more complex changes in the brain associated with the illness. While we do not fully understand how antidepressants work, we know that increasing monoamine neurotransmitters is an oversimplification. ("Monoamines" also needs to be the singluar "monoamine" in the sentence)
I also disagree with the first line of the abstract. It implies that no treatments are effective. Some treatments are effective in some patients. I also recommend replacing emotional with mental. Consider changing to something like this:
Major Depression disorder (MDD) is a potentially life-threatening mental illness, however many patients have a poor response to current treatments.
Author Response
This review provides an important update on neuro inflammatory mechanisms that contribute to the development of major depressive disorder (MDD), an illness that is a significant contributor to the overall global burden of disease. Work from the authors and others have made important advances in explaining how oxidative stress, inflammation, and alterations in the glymphatic system are involved in the pathophysiology of the illness. The review is well-organized and highlights the relevant structures, cells, molecules and mechanisms, with the figures providing additional explanations and clarifications. Below are the issues I recommend the authors address prior to publication:
Response: Thank you very much for the positive comments and constructive suggestions.
The primary issue with this paper is with English language and style. Extensive revisions are needed throughout the paper to improve English grammar and usage. I will provide some general and specific examples below, but most paragraphs need one or more revisions.
Inconsistent use of singular vs. pluaral
Present and past tense in the same sentence
"the" is missing in several sentences, for example:
THE Glymphatic system is a recently found structure….. (line 16)
Response: Thank you very much for the constructive suggestion, we have revised English language extensively by a native speaker and the MDPI publishing agency.
More and more possible mechanisms for MDD are emerging... (line 31) "More and more" is not appropriate
Too much use of "Indeed"
Do not use only the abbreviation MDD in the title (at least spell it out in the title).
Response: Thank you very much. We have carefully checked the wordings and specifically for the acronym usage, and have marked them in the revised paper. For example, we have changed “more and more”; we also removed most of the indeed, and also changed the title.
With regard to content I recommend reviewing sections of the manuscript that discusses clinical aspects of MDD and treatment. Sentences that describe the symptoms of depression could be more complete. As a psychiatrist reading this draft, my impression is that the authors don't fully understand the clinical presentation of the illness or what we currently understand about treatments. I suspect this is because of how the paper is written and the English language issues rather than any lack of knowledge on the part of the authors. For example, saying in line 369 that "MDD is a serious emotional disorder characterized by anhedonia, loss of energy" is incomplete. Revising this statement to include a more complete description of the symptoms of depression is needed.
Response: Thank you very much for the constructive suggestions, we have added more description of the symptoms of MDD, please have a look at Line 492 (with revisions):
“MDD is a serious emotional disorder characterized by anhedonia and a loss of energy as reported in the DSM-IV stipulation [113], which is typically show as “depressed mood” and “loss of interest”, as well as symptoms of fatigue, sleep disturbance, and sexual dysfunctions.”
Line 509 (with revisions): “The core symptoms involve loss of interest or pleasure, tiredness and lack of energy, reduced appetite and weight loss, as well as feeling of worthlessness or self-blame”.
I also disagree with the sentence in line 143 "Monoamine deficiency has been suggested to be the major reason for MDD, and most antidepressants are used to increase the monoamines neurotransmitters." This is an outdated explanation for the cause of MDD and needs to be revised to reflect the more complex changes in the brain associated with the illness. While we do not fully understand how antidepressants work, we know that increasing monoamine neurotransmitters is an oversimplification. ("Monoamines" also needs to be the singular "monoamine" in the sentence)
I also disagree with the first line of the abstract. It implies that no treatments are effective. Some treatments are effective in some patients. I also recommend replacing emotional with mental. Consider changing to something like this:
Major Depression disorder (MDD) is a potentially life-threatening mental illness, however many patients have a poor response to current treatments.
Response: Thank you very much for the professional suggestions, we have changed the description accordingly. For example, the first sentence of abstract:
“Major Depression disorder (MDD) is a potentially life-threatening mental illness, however many patients have a poor response to current treatments.”
Line 284 (with revisions) has been changed to:
“Monoamine deficiency was previously suggested as the major cause of MDD, and most antidepressants are used to increase the amount of monoamine neurotransmitters”.
Line 285 (with revisions) we have added the sentence:
“However, this might be an outdated explanation for the cause of MDD and needs to be revised to reflect the more complex changes in the brain associated with the illness. While we do not fully understand how antidepressants work, we know that increasing monoamine neurotransmitter is an oversimplification. For example, monoaminergic antidepressants are not working properly effective in some hard difficult-to-treat patients”
In addition, thanks for pointing out the plural format of "Monoamines", we have changed them into the singular "monoamine" format in the sentences.
Reviewer 3 Report
This review article entitled “Glymphatic dysfunction induced oxidase stress and neuro-inflammation in MDD” by Gu et al. touches on the issue of glymphatic system, a relatively new discovery in the brain function, and a possibility of an interesting link between its dysfunction and depression. While many details such as the causality between the two remain to be elucidated, the topic is quite inspiring and would attract readers of Antioxidants. However, the figures/captions used in this article require improvements as described below.
Figure 1.
Vascular should be indicated in the figure, and perivascular space should be mentioned in caption as “green area”. Neurotransmitter is drawn as black dots with no mention in the caption. Are those circles lined up at the presynaptic membrane meant to be synaptic vesicles? If so, synaptic vesicle zone does not typically look like that. What are those big yellow arrows?
Figure 2.
The caption indicates the diagram is about MDD, however, the diagram does not show MDD. Instead, the term “neurodegenerative disorders” is used. This is quite misleading. Neurodegenerative disorders are diseases due to progressive loss of neurons, such as Alzheimer’s disease and Parkinson’s disease. MDD is not one of them.
Figure 3.
This figure and its description in the text are confusing and then do not match well. In the text, it is supposed to be tri-color scheme, however, the figure shows gradients of two color. The background shows red-ish and blue-ish gradients, but two arrows with blue and yellow gradients do not seem to correspond to the background colors. According to the description in the text, this figure was adopted from the author’s previous publication (citation #72). However, this article does not show three color model. Also, if 5-HT and sleep enhances GS, should catecholamine exacerbate MDD by suppressing clearance by GS? Overall, this figure do not help readers to understand the mechanism of MDD.
Minor point:
L106: Tau makes tangles within neurons which causes problem. It is different from Ab that is in the extracellular space. Therefore, clearance of tau by GS probably won’t help the problem.
Author Response
This review article entitled “Glymphatic dysfunction induced oxidase stress and neuro-inflammation in MDD” by Gu et al. touches on the issue of glymphatic system, a relatively new discovery in the brain function, and a possibility of an interesting link between its dysfunction and depression. While many details such as the causality between the two remain to be elucidated, the topic is quite inspiring and would attract readers of Antioxidants. However, the figures/captions used in this article require improvements as described below.
Response: Thank you very much for the comments and the valuable suggestion, we have carefully checked our manuscript and changed accordingly.
Figure 1.
Vascular should be indicated in the figure, and perivascular space should be mentioned in caption as “green area”. Neurotransmitter is drawn as black dots with no mention in the caption. Are those circles lined up at the presynaptic membrane meant to be synaptic vesicles? If so, synaptic vesicle zone does not typically look like that. What are those big yellow arrows?
Response: Thanks a lot for the professional suggestions. We have added the descriptions for vascular, perivascular space, and neurotransmitters. We also changed synaptic vesicles in the presynaptic membrane. We also added some legends for glutamate vesicles, mGLuR, Na+ pump, Kir4.1 and AQP4.
Figure 2.
The caption indicates the diagram is about MDD, however, the diagram does not show MDD. Instead, the term “neurodegenerative disorders” is used. This is quite misleading. Neurodegenerative disorders are diseases due to progressive loss of neurons, such as Alzheimer’s disease and Parkinson’s disease. MDD is not one of them.
Response: Thanks a lot for pointing out the problem, we have changed “Neurodegenerative disorders” to “MDD” in Figure 2.
Figure 3.
This figure and its description in the text are confusing and then do not match well. In the text, it is supposed to be tri-color scheme, however, the figure shows gradients of two color. The background shows red-ish and blue-ish gradients, but two arrows with blue and yellow gradients do not seem to correspond to the background colors. According to the description in the text, this figure was adopted from the author’s previous publication (citation #72). However, this article does not show three color model. Also, if 5-HT and sleep enhances GS, should catecholamine exacerbate MDD by suppressing clearance by GS? Overall, this figure do not help readers to understand the mechanism of MDD.
Response: Thanks a lot for the important suggestions. We are very sorry for the confusing information, and we have changed colors in the figures accordingly.
The previous citation is 78-79, as below:
- Gu, S.; Wang, F.; Patel, N.P.; Bourgeois, J.A.; Huang, J. H. A model for basic emotions using observations of behavior in drosophila. Frontiers in Psychology. 2019, 10, 781. doi: 10.3389/fpsyg.2019.00781.
- Gu, S.; He, Z.; Xu, Q.; Dong, J.; Xiao, T.; Liang, F.; Ma, X.; Wang, F.; Huang, J.H. The relationship between 5-hydroxytryptamine and its metabolite changes with post-stroke depression. Frontier in Psychiatry. 2020, 13, 871754. doi: 10.3389/fpsyt.2022.871754.
Minor point:
L106: Tau makes tangles within neurons which causes problem. It is different from Ab that is in the extracellular space. Therefore, clearance of tau by GS probably won’t help the problem.
Response: Thanks a lot for the important suggestion. Even though most of tau makes tangles inside the neurons, they really present in the extracellular space as soluble materials:
(1).“During the past dozen years, a steadily accumulating body of evidence has indicated that soluble forms of Aβ and tau work together, independently of their accumulation into plaques and tangles, to drive healthy neurons into the diseased state and that hallmark toxic properties of Aβ require tau” Reference 1: Bloom GS. Amyloid-β and tau: the trigger and bullet in Alzheimer disease pathogenesis. JAMA Neurol. 2014 Apr;71(4):505-8. doi: 10.1001/jamaneurol.2013.584.
(2). “ tau may also be released from cells physiologically unrelated to protein aggregation. Tau secretion involves multiple vesicular and non-vesicle-mediated pathways, including secretion directly through the plasma membrane. Consequently, extracellular tau can be found in various forms, both as a free protein and in vesicles, such as exosomes and ectosomes. Once in the extracellular space, tau aggregates can be internalized by neighboring cells, both neurons and glial cells, via endocytic, pinocytic and phagocytic mechanisms.” Brunello CA, Merezhko M, Uronen RL, Huttunen HJ. Mechanisms of secretion and spreading of pathological tau protein. Cell Mol Life Sci. 2020 May;77(9):1721-1744. doi: 10.1007/s00018-019-03349-1.
(3). Sebastián-Serrano Á, de Diego-García L, Díaz-Hernández M. The Neurotoxic Role of Extracellular Tau Protein. Int J Mol Sci. 2018 Mar 27;19(4):998. doi: 10.3390/ijms19040998.
(4). Yamada K. Extracellular Tau and Its Potential Role in the Propagation of Tau Pathology. Front Neurosci. 2017 Nov 29;11:667. doi: 10.3389/fnins.2017.00667.
Reviewer 4 Report
The topic of this work is interesting even though the glymphatic system is widely described and reported in the literature. Perhaps, a little less evident in major depressive disorder.
In general, however, I have different major points to address:
1- The authors should better explain at the end of the introduction section the originality and the added value of their work and also how they want to structure their review. It is important to underline what is known in literature and what the authors want to add new and impactful prospective on this topic.
2- The different paragraphs should be joined together following a logical and connected thread, and I did not see these connections among the paragraphs.Furthermore, some paragraphs seem inserted in the work without any connection with the topic, for example that concerning the brain-gut axis and ROS.If the authors want to insert it, they must connect it to the rest of the topics in a more precise way ..... otherwise it must be removed.
3- Are the authors interested to Alzheimer Disease or to Major Depressive Disorder? This is confusing in my opinion. Major depressive disorder is not a neurodegenerative disorder tout court.
4- It seems to me that Figure 2 and Figure 4 depict the same situation in different ways … it could be better to draw one much better suggestive figure.
5- The conclusions should be much more incisive and they should look at future directions mainly in relation to translational level.
Minor points: check for some English errors and acronyms
Author Response
The topic of this work is interesting even though the glymphatic system is widely described and reported in the literature. Perhaps, a little less evident in major depressive disorder.
Response: Thanks a lot for the comments. We agree that there is less evidence for the effects of glymphatic system, there are really several papers about it. So this paper might provide a novel perspective for this field.
(1). Xia, M.; Yang, L.; Sun, G.; Qi, S.; Li, B. Mechanism of depression as a risk factor in the development of Alzheimer's disease: the function of AQP4 and the glymphatic system. Psychopharmacology (Berl). 2017, 234(3), 365–379.
(2). Wei et al. Chronic stress impairs the aquaporin-4-mediated glymphatic transport through glucocorticoid signaling. Psychopharmacology (Berl). 2019, 236(4): 1367-1384.
In general, however, I have different major points to address:
1-. The authors should better explain at the end of the introduction section the originality and the added value of their work and also how they want to structure their review. It is important to underline what is known in literature and what the authors want to add new and impactful prospective on this topic.
Response: Thank you very much, we have added a section in the introduction about the structure of the review.
Please see line 67 (with revisions): “In this paper, we reviewed recent studies about glymphatic system, and proposed a new perspective for its function in MDD. Firstly, we introduced the definition of glymphatic system and its major functions in clearance of wastes in the brain, including ROS, which might play a pivot role in clearance of ROS in MDD; secondly, we explored the roles of ROS in MDD, and introduced two major ways of modulating ROS by generation of ROS in mitochondria and anti-oxidant supplementation in the gut-brain axis and its mechanisms in inducing inflammation in microglia. Then we compared this process with the traditional monoamine hypothesis, and its relationship with the monoamines. Lastly, we reviewed the roles of two major glial cells in modulating the glymphatic systems and inflammation process. We hope this review will add new and impactful prospective for future studies of the glymphatic system and MDD, which might shed light in the mechanisms and treatment for MDD.”.
2- The different paragraphs should be joined together following a logical and connected thread, and I did not see these connections among the paragraphs. Furthermore, some paragraphs seem inserted in the work without any connection with the topic, for example that concerning the brain-gut axis and ROS.If the authors want to insert it, they must connect it to the rest of the topics in a more precise way ..... otherwise it must be removed.
Response: Thanks a lot for the important suggestion, we have added the links among different parts, for example:
(1) Line 104 (with revisions): “The glymphatic system is a unique drainage system that can drain fluid together with many other substances, such as macromolecules, ROS, cytokines, and antigens, from interstitial fluid to cerebrospinal fluid and finally to meningeal lymphatic vessels, to remove these materials to the peripheral lymphatic system [23]. The typical clearance materials include followings.”
(2) Line 183: “Dysfunction of glymphatic system can induce ROS accumulation and pro-inflammatory signaling, damage vital macromolecules, and induce cellular apoptosis during MDD”
(3) Line 201 (with revisions), For the brain-gut axis, “In addition to impaired ROS scavenging and excessive ROS generation, oxidative stress can also be affected by reduced antioxidant supply in the gut”.
(4) Line 401 (with revisions), “The reduced function coverage of blood vessels by astrocytic end feet to form theof astrocytic activity can impair the glymphatic system might which might be a major pathwayreason for for MDD [101]. The impairment of astrocyte includes many ion channels, such as Na+/Ca2+ exchanger, Na+/K+/2Cl-- cotransporter 1 as well as glial fibrillary acidic protein (GFAP) and aquaporin-4 (AQP4) , with a significant reduction in the coverage of gray matter vessels by AQP4 staining astrocyte processes was seen in MDD [102]. This suggests that alterations in AQP4 functions in astrocytes affect the regulation of water homeostasis, blood flow, glucose transport and metabolism, the blood–-brain barrier, glutamate turnover, and synaptic plasticity [103].”
(5) Line 442 (with revisions), we have removed this whole paragraph, in addition to many more sentences in the manuscript.
(6) Line 436-458 (with revisions), we have added a link.
Thank you very much for the valuable suggestion.
3- Are the authors interested to Alzheimer Disease or to Major Depressive Disorder? This is confusing in my opinion. Major depressive disorder is not a neurodegenerative disorder tout court.
Response: Thanks, we are sorry for the confusion, we have changed both Alzheimer Disease and neurodegenerative disorder to brain disorder.
4- It seems to me that Figure 2 and Figure 4 depict the same situation in different ways … it could be better to draw one much better suggestive figure.
Response: Thanks a lot for the valuable suggestion, we are sorry for the confusion. Figure 4 is actually about the contributions of microglia and inflammation to the glymphatic system. We have changed the legend of Figure 4 as:
“A model shows the contribution of microglial to the dysfunction of the glymphatic system, inducing ROS accumulation and inflammation in MDD. ROS accumulation and neuro-inflammation are the major reasons for MDD.”
5- The conclusions should be much more incisive and they should look at future directions mainly in relation to translational level.
Response: Thanks a lot, this is a very good point, we have curtailed the paragraph and changed the conclusion to include more perspectives for future studies in this field. Please see below:
“This review summarized recent studies about astrocytes, microglia, and inflammasome pathways that are activated during the occurrence and development of MDD, and provided a novel perspective for the mechanism of MDD. The perivascular space surrounding small blood vessels has recently been defined as the glymphatic system, providing a revolutionary perspective on the many pathophysiological mechanisms of MDD. Research into the function and pathogenesis of this system has proved that it carries out similar functions to the lymphatic system in the body, playing an important role in removing metabolic waste and maintaining homeostatic fluid circulation in the brain. In this article, we briefly described the factors influencing the cerebral glymphatic system, their effects on ROS accumulation and inflammation, and their roles in MDD. The aim of this research was to provide a perspective for future studies of the glymphatic system and MDD, and is expected to provide a new etiology and neural basis for the mechanisms and treatment for MDD.”
Minor points: check for some English errors and acronyms
Response: Thanks a lot for the suggestion, we have changed English language with both an English native speaker and also by the MDPI publishing agency.
Round 2
Reviewer 2 Report
The authors have largely addressed my comments from the original review. There are still a few minor English grammar edits needed (see the sentence at the end of the paragraph on lines 112-112), an occasional "the" missing.
I would also recommend revision of the sentence on line 409: Change DSMIV to DSM5, and delete the word "stipulation" - change it to "diagnostic critiera"
Other than these minor issues, I recommend publication.
Reviewer 3 Report
The authors responded to each of this reviewer's concern fairly well.
Reviewer 4 Report
The authors have answered all my questions and significantly improved the paper which is now suitable for publication.